# Antimicrobial and Antioxidant Properties of Bovine Livers and Hearts Hydrolysates

Ignė Juknienė [1,*], Gintarė Zaborskienė [1], Agnė Jankauskienė [1] and Irena Mačionienė [2]

1 Department of Food Safety and Quality, Lithuanian University of Health Sciences, Veterinary Academy, Tilzes St. 18, LT-47181 Kaunas, Lithuania; gintare.zaborskiene@lsmuni.lt (G.Z.); agne.jankauskiene@lsmuni.lt (A.J.)
2 Food Institute, Kaunas University of Technology, Radvilėnų pl. 19C-404, LT-50254 Kaunas, Lithuania; irena.macioniene@ktu.lt
* Correspondence: igne.jukniene@lsmu.lt; Tel.: +370-6811-4982

**Abstract:** Our previous research has indicated that bioactive protein hydrolysates derived from porcine by-products possess the potential to be utilized in the production of functional additives and food supplements. The objective of this investigation was to assess the antioxidant and antimicrobial characteristics and amino acid changes in hydrolysates of lyophilized meat of bovine livers and hearts. The relevant enzymes, papain and pepsin, were used to hydrolyze the meat by-products over periods of 3, 6, and 24 h. The antimicrobial properties of all enzymatically digested samples were assessed against *Listeria monocytogenes*, *Bacillus cereus*, *Salmonella enterica* subsp. *enterica* Serovar Typhimurium. *Bacillus cereus*, and *Escherichia coli*, *S. aureus* subsp. *aureus*. The assessment of antiradical activity involved the quantification of DPPH• and ABTS•+ absorbance in bovine by-product hydrolysates. The hydrolysates were subjected to amino acid analysis using AccQ Tag technology, which was performed by Waters Corporation in Milford, MA, USA. The bacteria *L. monocytogenes* had the highest antibacterial activity (inhibition zone) (20.00 ± 0.20 mm) and less against *E. coli* (10.00 ± 0.10 mm) of bovine heart hydrolysates and were prepared for 24 h with papain. The highest values of ABTS•+ (98.1 ± 0.30%) and of DPPH• scavenging activity (92.56 ± 0.56%) of cationic radicals were evaluated in the bovine liver hydrolysates after the effect of papain for 24 h. Longer hydrolysis time influenced the decrease in free hydrophobic amino acids (Ala, Val, Ile, Leu, Tyr, Phe, Pro, Met). The results confirmed the potential use of bovine liver and heart hydrolysates as functional or biologically active materials.

**Keywords:** by-products; hydrolysates; antioxidant activity; antimicrobial activity





## 1. Introduction

The optimization of meat by-products provides an opportunity to produce new components that could be used in food products or supplements [1,2]. Bovine meat by-products are rich in nutrients, especially proteins, which can have functional properties and be used for protein hydrolysis [3–5]. The generation of new peptides using enzymatic hydrolysis from meat by-products containing proteins provides the opportunity to obtain hydrolysates that may have antimicrobial and antioxidant properties [6–8]. In our opinion, the use of natural enzymes such as pepsin or papain in the production of meat by-products hydrolysates is an easier way in order to obtain desired antibacterial and antioxidant properties than the fermentation of proteinaceous raw materials with lactic acid bacteria or hydrolysis under the action of organic acids. Also, the use of the mentioned enzymes would be a sustainable way to obtain hydrolysates close to the human digestive system.

Protein enzymatic hydrolysis is a method for improving protein-containing products by obtaining biologically active amino acid sequences and improving functional or nutritional qualities [9]. Protein hydrolysates rich in small dipeptides and tripeptides with minimal free amino acids are common in the diet and have nutritional and therapeutic

value [10]. Enzymatic hydrolysis using papain and pepsin has been observed to produce the best results for the extraction of antioxidant and anti-inflammatory biopeptides from animal tissues [11].

Animal proteins are rich sources of important amino acids that are typically absent in plant proteins. These include methylhistidine and hydroxymethyllysine, which can be utilized to produce peptides with antioxidant characteristics [12]. The amino acid profile of animal liver is composed of aromatic (tyrosine and phenylalanine) and hydrophobic (leucine, valine, and isoleucine) amino acids, which exhibit high antioxidant activity [13]. Hearts are a protein source of biologically active components with hypolipidemic properties [14]. Thus, the animal liver or heart tissue is a raw material of high biological value, useful as a bioactive source for peptide hydrolysates with increased nutritional and functional value.

Isolation of new peptides from animal by-products with antioxidant properties can positively affect meat preservation, preventing lipid oxidation in food products and maintaining unchanged taste and smell [15,16]. Peptides possessing antioxidant characteristics have the potential to enhance nutritional value when employed as functional components or nutraceuticals [9]. The antioxidant activity and functional qualities of protein hydrolysates are influenced by several factors, such as the conditions under which hydrolysis occurs, the source of the protein, and the specific substrate proteins that are chosen. Protein hydrolysates may have advantages over purified peptides because of the production of oligopeptides due to absorption. Hydrolysates were also observed to have antioxidant activity comparable to purified peptides [17]. The antioxidant properties of peptides derived from meat products are attributed to their ability to inhibit the activity of reactive oxygen species (ROS) and lipid peroxidation and chelation of transition metal ions [18,19]. Peptides with antimicrobial properties isolated from animal by-products cover a smaller area of research compared to peptides with antioxidant properties [19]. Several peptides with antimicrobial properties were isolated from bovine hemoglobin proteins and bovine sarcoplasmic proteins [20,21], but there is a lack of data on the antibacterial properties of other bovine meat by-products proteins hydrolysates. The process of formation of bioactive peptides is a multifaceted mechanism that is influenced by numerous factors. In the process of enzymatic hydrolysis of proteins always new substances appear that determine the composition of the substrate. Those substances can chemically interact with each other and change the conditions of the medium and the activity of the existing enzymes. Therefore, in this experiment, we were interested in following how the hydrolysis time can change the amount of free amino acids and how the antibacterial and antioxidant activity of the hydrolysate will be changed.

The purpose of this study was to evaluate the antioxidant and antibacterial properties of lyophilized meat by-products from bovine livers and hearts.

## 2. Materials and Methods

Bovine livers ($n$ = 16) and hearts ($n$ = 16) were procured from slaughterhouses X and Y on one day. Samples were free of diseases, selected after post-slaughtering expertise, and subsequently transported to the laboratory within 24 h after the slaughtering in containers maintained at a temperature of 4 °C. The complete research scheme is presented in Figure 1.

Prior to the freeze-drying procedure, the bovine meat by-products were microscopically examined for secondary parasite screening, and parasite-free 8 of each type of offal were selected for further studies, sliced into pieces of 2 × 2 cm. Bovine livers ($n$ = 8) and hearts ($n$ = 8) samples were quickly frozen using a Liebherr freezer (LGv 5010 MediLine, Liebherr, Baden-Wurtemberg, Germany) at −35 °C for three hours. The freeze-drying (FD) procedure was conducted using a lyophilizer (Harvest Right, 84 North St, Salt Lake City, UT, USA) at a temperature of −80 °C and a vacuum pressure of 20 Pa for 72 h.

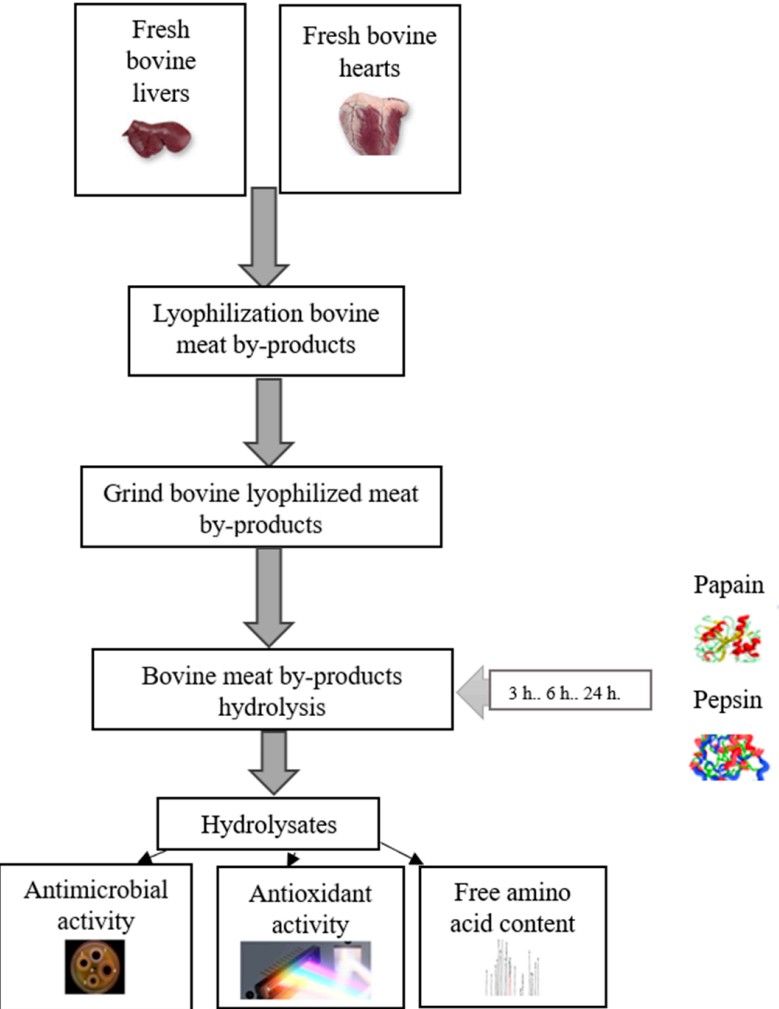

**Figure 1.** Diagram illustrating the complete research methodology.

*2.1. Preparation of Meat By-Products Hydrolysates*

Freeze-dried bovine livers (*n* = 8) and hearts (*n* = 8) were ground into powder using a laboratory-scale mill (Fritsch Grind Pulverisette 14, Indar-Oberstein, Germany) were used to mill the lyophilized bovine hearts and livers after which they were sieved through a 200 μm sieve (Figure 2). Enzymatic hydrolysis was performed using papain (Sigma-Aldrich, St. Louis, MO, USA) and pepsin (Sigma-Aldrich, St. Louis, MO, USA). Homogenization of selected animal by-products was accomplished through mixing with ice (1:1 lyophilized powder/ice). The optimal temperature and pH were chosen for the enzyme: papain (37 °C, pH = 6) and pepsin (37 °C, pH = 2.5). The pH of the hydrolysates was optimized using NaOH or HCl 1 N. In all incubation times, 3 h, 6 h, and 24 h, an enzyme-to-substrate ratio of 10 g/kg (*w*/*w*) was used. Following the incubation period, the hydrolysates were subjected to a heat treatment at a temperature of 95 °C in order to deactivate the enzymes. Subsequently, the hydrolysates were cooled using an ice bath. The hydrolysates underwent filtration using filter paper, followed by centrifugation at a speed of 4000 revolutions per minute for a duration of 10 min at a temperature of 4 °C. The upper phase of each hydrolysate was frozen at −80 °C until analysis.

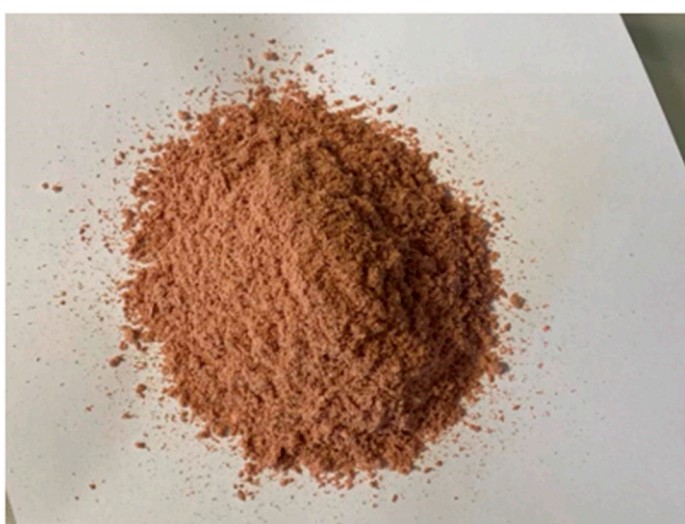

**Figure 2.** Bovine liver powder after lyophilization.

### 2.2. Free Amino Acid Composition of Bovine Livers and Hearts Hydrolysates

The hydrolysis of bovine meat by-products hydrolysates was carried out according to Commission Regulation No. 152/2009 [22]. Samples underwent hydrolysis, were cooled to room temperature, and then prepared. Then, 50 mL of L-2-aminobutyric acid at a concentration of 100 mmol/mL was added as an internal standard after a 100 mL volumetric flask was pipetted with the filtered hydrolysate. The contents of the flask were carefully combined, diluted to the appropriate concentration with ultrapure water, and then filtered using a 0.22 mm syringe filter. Amino acid derivatization was performed according to the Waters AccQ Tag Chemistry Kit instruction manual and was used to generate the derivatization reagents, calibration standards, and sample derivatization [23]. The Shimadzu low-pressure gradient HPLC system (Shimadzu Corp., Kyoto, Japan) was used to analyze the amino acids in the samples. This system includes the LC-10ATVP solvent delivery module, SIL-10ADVP auto-injector, CTO-10ACVP column oven, RF-10AXL spectrofluorometric detector, SCL-10AVP system controller, and DGU-14A online degasser. The Lab Solution LC system (Shimadzu Corp., Kyoto, Japan) was utilized to control the HPLC equipment and data collection. In this study, a Nova-Pak C18 chromatography column (4 mm, 150 3.9 mm; Waters Corp., Milford, MA, USA) was employed to achieve the separation of amino acid derivatives. The column was operated at a temperature of 37 °C, and a 10 mL injection of the derivative was introduced for analysis. At wavelengths between Ex 250 nm and Em 395 nm, separated derivatives were found. The gradient flow rate was set at 1.0 mL/min., which was used to separate the amino acid derivatives. In the mobile phase, eluent A was used. (made by mixing 1 L of ultrapure water with 100 mL of Waters AccQ Tag Eluent A Concentrate), eluent B (acetonitrile), and eluent C (ultrapure water).

The optimal separation of amino acid derivatives was attained with the implementation of the subsequent gradient program: The concentration of A was decreased from 100% to 98%, while the concentration of B was increased from 0% to 2% during the time interval of 0.0–0.5 min. Subsequently, the concentration of A was further decreased from 98% to 94%, while the concentration of B was increased from 2% to 6% during the time interval of 0.5–18.0 min. Following this, the concentration of A was decreased from 94% to 90%, while the concentration of B was increased from 6% to 10% during the time interval of 18.0–19.0 min. Moreover, the concentration of A was decreased from 90% to 83%, while the concentration of B was increased from 10% to 17% during the time interval of 19.0–29.5 min. Subsequently, the concentration of A was decreased from 83% to 0%, while the concentration of B was increased from 17% to 60%. Additionally, the concentration of C was increased from 0% to 40% during the time interval of 29.5–35.0 min. Following this, the

concentrations of B and C were held constant at 60% and 40%, respectively, for a duration of 5.0 min. Finally, a linear gradient was applied to the concentration of A, increasing it from 0% to 100% over a period of 1.0 min and then maintaining it at 100% for 10.0 min. The identification and quantification of single amino acids were achieved using a comparative analysis of the retention durations and peak areas of isolated amino acids in the sample in relation to the peak areas observed in the standard solution.

### 2.3. (DPPH•) Free Radical Scavenging Assay

The peptide fractions' ability to scavenge DPPH radicals (DPPH•) was examined in accordance with Brand-Williams et al. [24], with a minor change [25] in relation to the amounts of antioxidant solution used. A volume of 0.04 mL of the sample was combined with 0.96 mL of a 6 μM solution of DPPH• in 75% methanol. The measurement of absorbance was conducted three minutes after the initiation of the reaction at a wavelength of 515 nm, using the J.P. ELECTA S.A. V-1100D spectrophotometer from Barcelona, Spain. A solution of 75% methanol in redistilled water was employed for blank. The scavenging effect was calculated through the subsequent equation:

$$\text{Scavenging activity (\%)} = [1 - (A\text{ sample}/A\text{ control})] \times 100$$

### 2.4. ATBS•+ Radical Cation Scavenging Assay

ABTS radical cation was produced according to [26] by the reaction between 7 mM aqueous solution of ABTS in absolute ethanol and 2.45 mM potassium persulphate ($w/w$) during storage in a dark environment at room temperature for a duration of 12–16 h. Prior to use, the ABTS solution underwent dilution with absolute ethanol to obtain an absorbance reading of $0.8 \pm 0.03$ at a wavelength of 734 nm. A reagent blank reading was taken A control. A volume of 20 μL of meat by-product hydrolysate at different concentrations was introduced into 980 μL of ATBS solution and allowed to react for a duration of 6 min. The combination obtained was further analyzed using a J.P. SELECTA S.A. V-1100D spectrophotometer (Barcelona, Spain). The measurement of absorbance for the mixture was conducted at a designated wavelength of 734 nm.

The scavenging effect was calculated through the subsequent equation:

$$\text{Scavenging activity (\%)} = [1 - (A\text{ sample}/A\text{ control})] \times 100$$

### 2.5. The Ferric Reducing-Antioxidant Power (FRAP) Activity

The assessment of FRAP was according to the methodology [27]. In summary, 900 μL of recently prepared FRAP reagent was combined with 100 μL of the hydrolysate sample. The FRAP reagent consisted of a mixture of 300 mM acetate buffer (pH 3.6), 20 mM ferric chloride solution, and 10 mM TPTZ in a ratio of 10:1:1. The resulting mixture was incubated at 37 °C for 40 min, and the absorbance at 593 nm was measured using a J.P. SELECTA S.A. V-1100D spectrophotometer (Barcelona, Spain). The FRAP values were determined by comparing the change in absorption in the test mixture with that of increasing concentrations of $Fe^{3+}$ and were represented as millimoles of $Fe^{2+}$ equivalents per milliliter of sample. The standard curve was prepared using aqueous standard solutions of ferrous sulfate.

### 2.6. Determination of Antimicrobial Activity

The agar well diffusion method was employed to assess the antibacterial activity of lyophilized hydrolysates derived from bovine meat by-products. The cultures of reference strains *Listeria monocytogenes* ATCC 13932, *Bacillus cereus* ATCC 11778, Staphylococcus aureus subsp. aureus ATCC 25923, *Salmonella enterica* subsp. *enterica* serovar Typhimurium ATCC 14028 and *Escherichia coli* ATCC 25922 were pre-cultivated on PCA (Plate Count Agar, Liofilchem, Roseto degli Abruzzi Térano, Italy) slants at 37 °C or 30 °C overnight. The mature bacterial cultures were rinsed off the agar using a sterile physiological saline solution composed of 0.85% sodium chloride in distilled water. The cell suspension of each

culture was adjusted in accordance with the McFarland standard No. 0.5. Then, 1 mL of the prepared suspension was added into 100 mL of melted and cooled to 45 °C PCA medium. The prepared mixture of bacteria cell suspension and the medium was mixed thoroughly and poured into Petri dishes (90 mm), 12 mL each. After the medium had solidified, wells of 8 mm diameter were made in the plates, which were filled with 50 μL of the examined solution. The antimicrobial activity of lyophilized meat by-product hydrolysates against the pathogenic bacteria cultures was evaluated after 24 h of growth at 37 °C or 30 °C. The measurement of the diameter of inhibitory zones was conducted using calipers with a precision of 0.5 mm. Distilled water was employed as the control in the blank sample.

*2.7. Determination of Minimum Inhibitory Concentration (MIC)*

The minimum inhibitory concentration (MIC) of lyophilized and hydrolyzed meat by-products with papain was determined using the broth microdilution method. In the test, samples were added to Tryptic soy broth (Liofilchem diagnostic, Roseto degli Abruzzi, Italy) to obtain a final concentration of 10 μg/mL, which were further serially diluted to 5.0, 2.5, 1.25, 0.0625, 0.0313, 0.0156, 0.0078, and 0.0039 μg/mL, respectively. A total of 50 μL *L. monocytogenes* culture suspension (approximately $1.5 \times 10^8$ CFU/mL) was transferred into each tube. All tubes, including the control, containing the bacterial suspension in Tryptic soy broth were incubated at $37 \pm 1$ °C for 24 h. The lowest concentrations of the test samples, which, after macroscopic evaluation, did not show any growth of *L. monocytogenes* culture, were determined as MICs and were expressed in μg/mL. All tests were performed in triplicate. Due to thesmall inhibition zones shown, the minimum inhibitory concentrations (MICs) of tested hydrolysates (after 24 h with papain) against *S. enterica* subsp. *enterica serovar Typhimurium* were not evaluated.

*2.8. Statistics*

The standard deviation (SD) of each determination's mean value is used to report the experimental results. The obtained results were analyzed by SPSS for Windows, version 25.0 (IBM SPSS Inc., Chicago, IL, USA, 2017). Bovine hydrolysate hearts ($n = 8$) and livers ($n = 8$), each with three duplicates ($n = 3$ duplicates), were used in the study ANOVA, and the *p*-value at $p < 0.05$ was considered significant. Statistically significant differences ($p < 0.05$) among the hydrolysate treatments were determined using Duncan's multiple-range test.

**3. Results and Discussion**
*Amino Acids*

The quantity of free amino acids in the hydrolysates of bovine liver and hearts showed significant differences at different times of hydrolysis (Table 1). The composition of amino acid profiles in hydrolysates consists of both free amino acids and short-chain peptides [28]. The levels of free amino acids in the by-products of bovine meat by-products exhibited the lowest values following a 24-h incubation treatment time with papain and pepsin enzymes. The amounts of hydrophobic amino acids (Ala, Val, Ile, Leu, Tyr, Phe, Pro, Met) tend to decrease with increasing hydrolysis time. María López-Pedrouso et al. reported similar findings, indicating that an increase in hydrolysis time resulted in a decrease in the levels of free amino acids [13]. In our study, papain was the most effective in reducing free amino acids in both hydrolysates: heart and livers. According to Anzani et al., papain is the most effective enzyme at hydrolyzing meat proteins rich in glycine and hydroxyproline, while other enzymes, such as trypsin and pancreatin, are less effective in dissolving meat, producing hydrolysates with less glycine and hydroxyproline [29]. It has been observed that the hydrophobic amino acids Gly and Pro can enhance antioxidant activity due to increased lipid solubility and free radical reactions [30]. Furthermore, it has been observed that certain amino acids, including Lysine (Lys), Tyrosine (Tyr), and Histidine (His), possess inherent antioxidant capabilities when present in peptide structures. In addition, Cysteine (Cys) has proton-donating properties, while essential amino acids

can chelate metal ions [31]. Considering the outcomes that have been obtained, we may conclude that bioactive peptides can also be formed from free amino acids, connecting them to specific sequences. Some amino acids, particularly sulfur-containing amino acids like Cys, can act as antioxidants themselves, so the resulting peptide sequences with these amino acids in extremely short chains can have an excellent antioxidant effect.

**Table 1.** Free amino acid content of the hydrolyzed bovine livers and hearts g/100 g.

| Amino Acids | Time (h) | Bovine Liver | | Bovine Heart | |
| | | Pepsin | Papain | Pepsin | Papain |
|---|---|---|---|---|---|
| Ala | 3 | $0.31 \pm 0.10$ | $0.26 \pm 0.30$ | $0.34 \pm 0.02$ | $0.28 \pm 0.02$ |
| | 6 | $0.27 \pm 0.02$ | $0.22 \pm 0.03$ | $0.28 \pm 0.10$ | $0.24 \pm 0.03$ |
| | 24 | $0.25 \pm 0.10$ | $0.18 \pm 0.09$ | $0.24 \pm 0.12$ | $0.16 \pm 0.02$ |
| | *p-value (Time)* | * | * | * | * |
| Thr | 3 | $0.12 \pm 0.03$ | $0.11 \pm 0.03$ | $0.18 \pm 0.08$ | $0.16 \pm 0.03$ |
| | 6 | $0.09 \pm 0.03$ | $0.07 \pm 0.08$ | $0.14 \pm 0.03$ | $0.15 \pm 0.05$ |
| | 24 | $0.05 \pm 0.01$ | $0.04 \pm 0.13$ | $0.06 \pm 0.01$ | $0.12 \pm 0.06$ |
| | *p-value (Time)* | * | * | * | * |
| Ser | 3 | $0.15 \pm 0.08$ | $0.12 \pm 0.09$ | $0.17 \pm 0.01$ | $0.15 \pm 0.02$ |
| | 6 | $0.11 \pm 0.05$ | $0.09 \pm 0.01$ | $0.13 \pm 0.02$ | $0.16 \pm 0.01$ |
| | 24 | $0.08 \pm 0.3$ | $0.05 \pm 0.13$ | $0.03 \pm 0.01$ | $0.15 \pm 0.02$ |
| | *p-value (Time)* | * | * | * | |
| Glu | 3 | $0.62 \pm 0.11$ | $0.58 \pm 0.18$ | $0.58 \pm 0.4$ | $0.45 \pm 0.13$ |
| | 6 | $0.58 \pm 0.13$ | $0.56 \pm 0.12$ | $0.54 \pm 0.3$ | $0.38 \pm 0.12$ |
| | 24 | $0.48 \pm 0.11$ | $0.43 \pm 0.13$ | $0.44 \pm 0.12$ | $0.28 \pm 0.11$ |
| | *p-value (Time)* | * | * | * | * |
| Pro | 3 | $0.19 \pm 0.04$ | $0.17 \pm 0.02$ | $0.21 \pm 0.03$ | $0.14 \pm 0.03$ |
| | 6 | $0.17 \pm 0.01$ | $0.15 \pm 0.01$ | $0.15 \pm 0.01$ | $0.9 \pm 0.01$ |
| | 24 | $0.12 \pm 0.04$ | $0.11 \pm 0.06$ | $0.10 \pm 0.01$ | $0.3 \pm 0.01$ |
| | *p-value (Time)* | * | * | * | * |
| Gli | 3 | $0.22 \pm 0.01$ | $0.24 \pm 0.02$ | $0.25 \pm 0.02$ | $0.21 \pm 0.12$ |
| | 6 | $0.14 \pm 0.02$ | $0.11 \pm 0.01$ | $0.18 \pm 0.02$ | $0.18 \pm 0.02$ |
| | 24 | $0.11 \pm 0.06$ | $0.08 \pm 0.01$ | $0.08 \pm 0.02$ | $0.14 \pm 0.04$ |
| | *p-value (Time)* | * | * | * | * |
| Ala | 3 | $0.27 \pm 0.01$ | $0.29 \pm 0.02$ | $0.32 \pm 0.08$ | $0.27 \pm 0.10$ |
| | 6 | $0.22 \pm 0.02$ | $0.27 \pm 0.01$ | $0.25 \pm 0.01$ | $0.25 \pm 0.02$ |
| | 24 | $0.19 \pm 0.01$ | $0.09 \pm 0.01$ | $0.18 \pm 0.03$ | $0.22 \pm 0.01$ |
| | *p-value (Time)* | * | * | * | * |
| Val | 3 | $0.19 \pm 0.02$ | $0.10 \pm 0.01$ | $0.21 \pm 0.03$ | $0.27 \pm 0.01$ |
| | 6 | $0.17 \pm 0.04$ | $0.12 \pm 0.02$ | $0.22 \pm 0.02$ | $0.25 \pm 0.02$ |
| | 24 | $0.13 \pm 0.03$ | $0.09 \pm 0.02$ | $0.15 \pm 0.01$ | $0.22 \pm 0.01$ |
| | *p-value (Time)* | * | * | * | * |
| Met | 3 | $0.13 \pm 0.06$ | $0.10 \pm 0.02$ | $0.11 \pm 0.01$ | $0.15 \pm 0.05$ |
| | 6 | $0.14 \pm 0.07$ | $0.12 \pm 0.04$ | $0.08 \pm 0.01$ | $0.13 \pm 0.04$ |
| | 24 | $0.10 \pm 0.01$ | $0.08 \pm 0.02$ | $0.04 \pm 0.01$ | $0.8 \pm 0.01$ |
| | *p-value (Time)* | * | * | * | * |
| Ile | 3 | $0.15 \pm 0.01$ | $0.17 \pm 0.02$ | $0.12 \pm 0.02$ | $0.21 \pm 0.03$ |
| | 6 | $0.17 \pm 0.02$ | $0.11 \pm 0.01$ | $0.15 \pm 0.01$ | $0.18 \pm 0.01$ |
| | 24 | $0.11 \pm 0.02$ | $0.09 \pm 0.02$ | $0.05 \pm 0.01$ | $0.12 \pm 0.01$ |
| | *p-value (Time)* | | * | * | * |
| Leu | 3 | $0.29 \pm 0.02$ | $0.22 \pm 0.1$ | $0.24 \pm 0.4$ | $0.25 \pm 0.05$ |
| | 6 | $0.31 \pm 0.02$ | $0.18 \pm 0.2$ | $0.25 \pm 0.02$ | $0.22 \pm 0.03$ |
| | 24 | $0.23 \pm 0.03$ | $0.17 \pm 0.02$ | $0.19 \pm 0.02$ | $0.18 \pm 0.02$ |
| | *p-value (Time)* | | | | |

**Table 1.** *Cont.*

| | | Bovine Liver | | Bovine Heart | |
|---|---|---|---|---|---|
| Amino Acids | Time (h) | Pepsin | Papain | Pepsin | Papain |
| Tyr | 3 | 0.09 ± 0.04 | 0.03 ± 0.01 | 0.08 ± 0.03 | 0.05 ± 0.01 |
| | 6 | 0.04 ± 0.03 | 0.02 ± 0.01 | 0.07 ± 0.01 | 0.03 ± 0.01 |
| | 24 | 0.01 ± 0.01 | 0.02 ± 0.01 | 0.02 ± 0.03 | 0.02 ± 0.01 |
| | *p-value (Time)* | * | | | * |
| Phe | 3 | 0.13 ± 0.02 | 0.11 ± 0.01 | 0.12 ± 0.03 | 0.15 ± 0.04 |
| | 6 | 0.11 ± 0.04 | 0.10 ± 0.09 | 0.08 ± 0.01 | 0.13 ± 0.03 |
| | 24 | 0.09 ± 0.04 | 0.02 ± 0.01 | 0.05 ± 0.01 | 0.02 ± 0.01 |
| | *p-value (Time)* | * | | * | * |
| His | 3 | 0.14 ± 0.02 | 0.11 ± 0.03 | 0.11 ± 0.02 | 0.17 ± 0.03 |
| | 6 | 0.11 ± 0.01 | 0.09 ± 0.01 | 0.08 ± 0.01 | 0.16 ± 0.02 |
| | 24 | 0.07 ± 0.01 | 0.05 ± 0.02 | 0.05 ± 0.01 | 0.05 ± 0.01 |
| | *p-value (Time)* | * | * | * | * |
| Lys | 3 | 0.32 ± 0.14 | 0.34 ± 0.13 | 0.38 ± 0.06 | 0.42 ± 0.15 |
| | 6 | 0.22 ± 0.13 | 0.27 ± 0.04 | 0.33 ± 0.08 | 0.38 ± 0.11 |
| | 24 | 0.19 ± 0.05 | 0.15 ± 0.04 | 0.18 ± 0.02 | 0.27 ± 0.02 |
| | *p-value (Time)* | * | * | * | * |
| Arg | 3 | 0.28 ± 0.02 | 0.25 ± 0.04 | 0.31 ± 0.05 | 0.31 ± 0.02 |
| | 6 | 0.22 ± 0.01 | 0.20 ± 0.05 | 0.25 ± 0.08 | 0.28 ± 0.05 |
| | 24 | 0.21 ± 0.06 | 0.19 ± 0.03 | 0.19 ± 0.03 | 0.23 ± 0.03 |
| | *p-value (Time)* | | * | * | |

Time effect between hydrolysates of bovine livers and heart; $p > 0.05$; * $p < 0.05$.

Three methodologies, DPPH● and ABTS●+ radical scavenging activity and iron-reducing antioxidant power (FRAP), were applied to evaluate the antioxidant activity of specific samples, as outlined in Table 2. The antioxidant activity values of bovine heart and liver hydrolysates were highest after 24 h ($p < 0.05$). The highest values of antioxidant activity values were determined ABTS●+ (98.10 ± 1.76%) and DPPH (92.56 ± 0.56%) by hydrolyzing bovine liver with papain for 24 h ($p < 0.05$). In addition, it was observed that bovine liver hydrolysates exhibited better antioxidant activity compared with heart hydrolysates. Other authors revealed that liver hydrolysates have a higher antioxidant capacity compared to other tissues, including the colon, pancreas, and appendix. In this regard, papain treatment resulted in higher antioxidant activity in bovine hearts and livers and was more effective than pepsin enzyme [32]. Also, the hydrolysis time of 24 h increased the free radical scavenging activity of DPPH● and ABTS●+. Immune system performance is generally enhanced by substances with antioxidant characteristics [33]. It is thus a very helpful method to evaluate the antioxidant potential of protein hydrolysates [34]. Among the bovine hydrolyzed samples, the highest FRAP value was measured in hydrolyzed bovine hearts (18.47 ± 1.22) and similar–in bovine livers (17.88 ± 0.98) after 24 h of hydrolysis with papain enzyme. FRAP values varied significantly ($p < 0.05$) depending on the duration of treatment. Prolonged hydrolysis periods enhanced the FRAP activity of the peptides generated during enzymatic hydrolysis. An increase in FRAP activity with increasing hydrolysis time was also recorded in various protein hydrolysates obtained from meat by-products [11]. Liu et al. found that using the papain enzyme and red blood cells as the substrate, FRAP activity increased in proportion to the duration of hydrolysis [35].

**Table 2.** Antioxidant activity [ATBS %, DPPH %, and FRAP (mM equivalent to $FeSO_4 \cdot 7H_2O$)] after hydrolysis using pepsin and papain of lyophilized bovine livers and hearts.

| Material | Enzymes | Time (h) | ATBS•+ Scavenging Activity | DPPH• Scavenging Activity | FRAP (mM Equivalent to $FeSO_4 \cdot 7H_2O$) |
|---|---|---|---|---|---|
| Bovine hearts | Pepsin | 24 | 68.22 ± 1.67 [a] | 46.62 ± 1.48 [a] | 16.45 ± 0.65 [a] |
| | | 6 | 67.22 ± 1.87 [a] | 45.23 ± 1.23 [a] | 13.23 ± 1.11 [b] |
| | | 3 | 64.61 ± 1.74 [a] | 10.02 ± 1.56 [b] | 9.22 ± 0.78 [c] |
| | Papain | 24 | 76.34 ± 0.98 [b] | 78.25 ± 1.13 [a] | 18.47 ± 1.22 [a] |
| | | 6 | 68.82 ± 1.56 [a] | 44.17 ± 1.88 [b] | 14.56 ± 0.95 [b] |
| | | 3 | 65.35 ± 1.11 [a] | 21.24 ± 1.46 [c] | 9.23 ± 1.28 [c] |
| Bovine livers | Pepsin | 24 | 84.81 ± 1.11 [c] | 96.12 ± 1.11 [a] | 16.98 ± 0.49 [a] |
| | | 6 | 65.30 ± 1.85 [a] | 38.82 ± 1.66 [b] | 13.11 ± 0.52 [b] |
| | | 3 | 64.92 ± 1.89 [a] | 13.06 ± 1.67 [c] | 9.22 ± 0.78 [c] |
| | Papain | 24 | 98.10 ± 0.30 [c] | 92.56 ± 0.56 [a] | 17.88 ± 0.98 [a] |
| | | 6 | 73.03 ± 0.91 [b] | 43.36 ± 0.67 [b] | 15.38 ± 0.68 [b] |
| | | 3 | 72.20 ± 0.88 [b] | 21.01 ± 0.72 [c] | 11.34 ± 0.87 [c] |

[a–c] Means in a column (in each group separately) a common superscript letter differs ($p < 0.05$) as analyzed by three- way repeated measures ANOVA test.

Peptides make up the majority of hydrolysates, which is what gives them their antioxidant capacity because they can also contain endogenous substances like ascorbic acid, which could boost the total antioxidant capacity [32]. Furthermore, the antioxidant activity seen in the hydrolysates can be attributed to alterations in the structural proteins that occur as a result of enzymatic hydrolysis. This phenomenon can be attributed to the capacity of amino acids to engage in interactions with oxidizing agents, resulting in modifications and subsequent consequences for the residues [36]. Data from previous studies show that more antioxidant peptides are formed when the hydrolysis time is longer, and this has significantly higher DPPH scavenging activity [32]. On the other hand, the results of our study show that as the hydrolysis time increases, the amounts of all tested free amino acids tend to decrease, and they are inversely correlated with the antioxidant activity of hydrolysates of bovine hearts and livers, regardless of the enzyme used, but the strongest inverse correlation was established in a case of bovine hearts hydrolysates with papain (r = 0.856, $p < 0.001$).

Antibacterial activity was determined using the agar diffusion assay (inhibition zone assay) (Table 3). The bovine heart hydrolysates exhibited the most significant antibacterial effects against *L. monocytogenes* (20.0 ± 0.2 mm), with considerable inhibition observed against *E. coli* (10.0 ± 0.1 mm) and *S. enterica* subsp. *enterica* serovar Typhimurium (10.0 ± 0.1 mm) after 24 h of hydrolysis with papain. Bovine livers hydrolysates with papain 24 h. also exhibited stronger antibacterial efficacy against *L. monocytogenes* (17.5 ± 0.2 mm), *S. enterica* subsp. *enterica serovar Typhimurium* (10.0 ± 0.1 mm). Nevertheless, it was discovered that bovine heart and liver hydrolysates treated with pepsin enzyme only had a bacteriostatic impact on *S. aureus* subsp. *aureus*. *L. monocytogenes* strain was assessed by determining the minimum inhibitory concentration. Lyophilized and hydrolyzed bovine heart and bovine liver with papain (after 24 h) exhibited strong antibacterial activity against *L. monocytogenes* growth (Figure 3). The minimum inhibitory concentration value was 1.25 μg/mL in both cases of bovine liver and heart hydrolysates. Antimicrobial activity may have been influenced by the size of the resulting peptides [37]. Kim and Wijesekara have reported that antibacterial activity can be observed in low molecular weight peptides (less than 10 kDa) that possess a positive charge and are made of amphipathic molecules [38,39]. In general, the hydrolysates of meat by-products source antibacterial properties that may result from either the synthesis of tiny peptides by the hydrolysis of proteins or the bacteriocins by lactic acid bacteria [17]. Akhilesh et al. observed that papain porcine liver hydrolysates showed the highest antibacterial activity after 6 h compared to

shorter hydrolyzation times [40]. Similar studies have concluded that the peptides in the hydrolysate determine the antibacterial activity rather than the chosen hydrolysis method [31]. In our studies, the antibacterial activity against *L. monocytogenes* growth of bovine liver and heart hydrolysates under the influence of papain during 24 h was inversely correlated with the amounts of Glu, His, and Lys, and moderately strong linear relationships were found when $p < 0.05$. Thus, our determined antibacterial activity against selected bacteria depended on the selected enzyme and hydrolysis time, and this resulted in the utilization of certain amino acids for the formation of specific peptides. More detailed studies and analysis of peptide sequences could reveal the application possibilities of bovine liver and heart hydrolysates with antibacterial activity against certain bacteria.

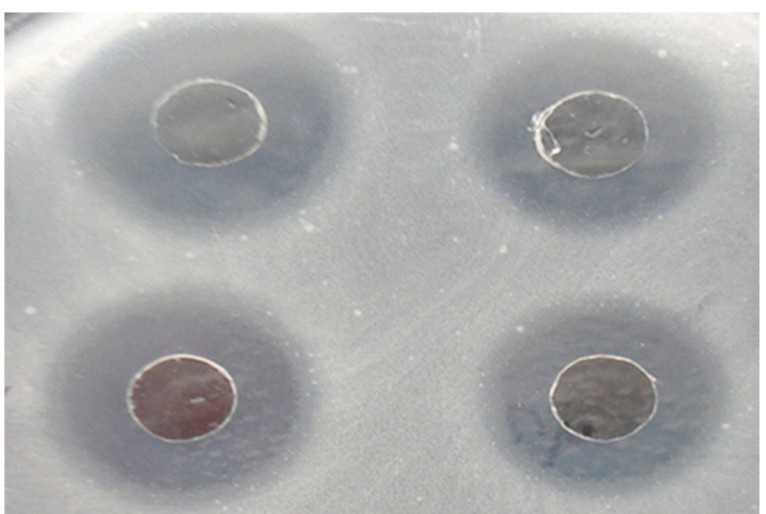

**Figure 3.** Antimicrobial activity after hydrolysis using pepsin of lyophilized bovine livers against *Listeria monocytogenes* ATCC 13932 growth.

**Table 3.** Antimicrobial activity after hydrolysis using pepsin and papain of lyophilized bovine livers and hearts.

| Material | Enzyme | Time (h) | Zone of Inhibition Diameter, mm | | | | |
|---|---|---|---|---|---|---|---|
| | | | *E. coli* ATCC 25922 | *S. aureus* subsp. *aureus* ATCC 25923 | *L. monocytogenes* ATCC 13932 | *S. enterica* subsp. *enterica* Serovar Typhimurium ATCC 14028 | *B. cereus* ATCC 11778 |
| Bovine heart | Pepsin | 24 | N.D. | bacteriostatic effect | N.D. | N.D. | $0 \pm 0$ |
| | | 6 | N.D. | $0 \pm 0$ | N.D. | N.D. | $0 \pm 0$ |
| | | 3 | N.D. | $0 \pm 0$ | N.D. | $0 \pm 0$ | $0 \pm 0$ |
| Bovine liver | Pepsin | 24 | N.D. | bacteriostatic effect | N.D. | $0 \pm 0$ | $0 \pm 0$ |
| | | 6 | N.D. | $0 \pm 0$ | N.D. | $0 \pm 0$ | $0 \pm 0$ |
| | | 3 | N.D. | $0 \pm 0$ | N.D. | $0 \pm 0$ | $0 \pm 0$ |
| Bovine heart | Papain | 24 | $10.0 \pm 0.1$ | bacteriostatic effect | $20.0 \pm 0.2$ | $10.0 \pm 0.1$ | $0 \pm 0$ |
| | | 6 | N.D. | bacteriostatic effect | $12.5 \pm 0.1$ | $0 \pm 0$ | $0 \pm 0$ |
| | | 3 | N.D. | bacteriostatic effect | $10.2 \pm 0.1$ | $0 \pm 0$ | $0 \pm 0$ |

**Table 3.** *Cont.*

| Material | Enzyme | Time (h) | E. coli ATCC 25922 | S. aureus subsp. aureus ATCC 25923 | L. monocytogenes ATCC 13932 | S. enterica subsp. enterica Serovar Typhimurium ATCC 14028 | B. cereus ATCC 11778 |
|---|---|---|---|---|---|---|---|
| | | | **Zone of Inhibition Diameter, mm** | | | | |
| Bovine liver | Papain | 24 | N.D. | bacteriostatic effect | 17.5 ± 0.2 | 10.0 ± 0.1 | 0 ± 0 |
| | | 6 | N.D. | bacteriostatic effect | 11.7 ± 0.2 | 10.0 ± 0.1 | 0 ± 0 |
| | | 3 | N.D. | bacteriostatic effect | 9.8 ± 0,1 | 10.0 ± 0.1 | 0 ± 0 |

N.D.—not detected.

## 4. Conclusions

The results showed the potential application of enzymatic hydrolysis in bovine by-products as a protein source in the generating of hydrolysates with antioxidant and antimicrobial effects. Antioxidant activity in hydrolysates of bovine hearts and livers is inversely correlated to the amount of free amino acids and increased with lengthening hydrolysis time. The bovine liver and heart hydrolysates, treated with papain for a duration of 24 h, exhibited the best antimicrobial and antioxidant properties. Hydrolysates from the bovine slaughter by-products offer the potential for the creation of value-added products. These products possess the potential for utilization in the development of functional food additives and bioactive substances for the production of dietary supplements. The application of antioxidant hydrolysates has the potential to enhance nutritional outcomes by aiding the body in mitigating the detrimental impacts of oxidative stress and diminishing the likelihood of degenerative diseases. Tested hydrolysates of lyophilized bovine heart and liver meat (after 24 h papain fermentation) can be used as a new alternative in the production of more sustainable disinfectants or medicines as substances with antimicrobial properties.

**Author Contributions:** Conceptualization, G.Z. and I.J.; methodology G.Z., I.J. and I.M.; software, A.J.; validation I.J.; formal analysis, G.Z.; investigation, I.J. and G.Z.; resources, I.J.; data curation, G.Z.; writing—original draft preparation, I.J. and A.J.; writing—review and editing, I.J. and I.M.; visualization, A.J. and I.J.; supervision, G.Z.; project administration, I.J. All authors have read and agreed to the published version of the manuscript.

**Funding:** This research received no external funding.

**Institutional Review Board Statement:** Not applicable.

**Informed Consent Statement:** Not applicable.

**Data Availability Statement:** The data presented in this study are available on request from the corresponding author. The data are not publicly available due to privacy.

**Conflicts of Interest:** The authors declare no conflict of interest.

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
