# Peer review of "Antimicrobial and Antioxidant Properties of Bovine Livers and Hearts Hydrolysates"

_applsci, doi:10.3390/app132413142_

Round 1

Reviewer 1 Report

Comments and Suggestions for Authors

Line 82: please mention the number of bovine livers and hearts collected.

Line 82: all bovine livers and hearts were collected in only one day? or were they collected in different days?

Line 82: Please mention if the bovine livers and hearts collected were free of diseases and parasites, at least visually.

Line201: What was the experimental design for free amino acid content, antioxidant activity, and antimicrobial activity? A reader with knowledge of statistics could infer, but it would be better if the authors explained it in depth for a wider audience.

Lines 201-207: Additionally, to the ANOVA analysis, it would be nice if the authors evaluated possible linear or quadratic effects due to time for free amino acid content, antioxidant activity, and antimicrobial activity.

Line 203: please be specific, eight samples of bovine hydrolysates of hearts and, eight samples of bovine hydrolysates of livers?

Line 204: were used in the study were considered for ANOVA analysis.

Line 231: Incubation Treatment time impact effect

Comments on the Quality of English Language

 Minor editing of English language required

Author Response

Dear, Reviewer,

First of all we want to thank you for your comments which are very well-aimed. 

Line 82: please mention the number of bovine livers and hearts collected.

We have added. Marking changed places in the manuscript in yellow.

Line 82: all bovine livers and hearts were collected in only one day? or were they collected in different days?

Bovine livers (n = 16) and hearts (n = 16) were procured from slaughterhouses X and Y on one day. Slaughterhouses were selected based on the number of animals slaughtered. X slaughterhouse slaughters up to 200 units per day, Y slaughterhouse up to 10.

Line 82: Please mention if the bovine livers and hearts collected were free of diseases and parasites, at least visually.

Bovine livers (n = 16) and hearts (n = 16) were procured from slaughterhouses X and Y on one day. Samples were free of diseases, selected after post slaughtering expertise. Prior the freeze-drying procedure, the bovine meat by-products were microscopically examined for secondary parasite screening and parasite-free 8 of each type of offal were selected for further studies, sliced into pieces of 2x2 cm. Bovine livers (n = 8) and hearts (n = 8) samples were quickly frozen.

Line201: What was the experimental design for free amino acid content, antioxidant activity, and antimicrobial activity? A reader with knowledge of statistics could infer, but it would be better if the authors explained it in depth for a wider audience.

In our experiment, as the time of enzymatic fermentation increased, the amount of free amino acids decreased, but the antioxidant and antibacterial activity of the hydrolysates increased. We can assume that bioactive peptides can also be formed from free amino acids, connecting them to specific sequences. Some amino acids, particularly sulfur-containing amino acids like cysteine ​​and methionine, can act as antioxidants themselves, so the resulting peptide sequences with these amino acids in extremely short chains can have an excellent antioxidant effect.

Lines 201-207: Additionally, to the ANOVA analysis, it would be nice if the authors evaluated possible linear or quadratic effects due to time for free amino acid content, antioxidant activity, and antimicrobial activity.

Thanks for the suggestion, as we will be using a lot of samples to review statistics in future articles

Line 203: please be specific, eight samples of bovine hydrolysates of hearts and, eight samples of bovine hydrolysates of livers?

We have added. Marking changed places in the manuscript in yellow.

Line 204: were used in the study were considered for ANOVA analysis.

We corrected.

Line 231: Incubation Treatment time impact effect

We corrected.

Reviewer 2 Report

Comments and Suggestions for Authors

MS is interesting but needs modifications

1. Introduction should be more clear, what is the novelty of the MS?

2. Line 82: details of X and Y and the method of sampling should be disclosed in the MS.

3. FRAP is a very important antioxidant measurement technique, it needs to be performed see https://doi.org/10.1111/jfpp.15233 for protocol and significance.

4. A schematic diagram of the experimental process needs to be included.

5. Why MIC is not done?

6. PCA is required for statistical analysis see https://doi.org/10.1002/fsn3.576

7. Pics for samples need to be included.

Comments on the Quality of English Language

Minor editing of the English language required

Author Response

Dear Reviewer,

First of all we want to thank you for your comments which are very well-aimed. We are attaching our answers.

Reviewer 3 Report

Comments and Suggestions for Authors

In this study, the authors evaluated the antioxidant and antimicrobial characteristics and observed changes in amino acid composition in hydrolysates of lyophilized bovine liver and heart tissues. The results showed the potential application of enzymatic hydrolysis in utilizing bovine by-products as a protein source for producing hydrolysates with antioxidant and antimicrobial properties. Here are my comments:

1.      The section title 'Results' should be revised to 'Results and Discussion.'

2.      It is noted that certain amino acids (tyrosine, phenylalanine, leucine, valine, and isoleucine), known for their high antioxidant activity, decreased with increased hydrolysis time. However, the antioxidant activity actually increased with extended hydrolysis time. The authors should discuss this apparent contradiction in the Results and Discussion section.

3.      The differences in results obtained when using pepsin and papain, as well as between liver and heart tissues, should be summarized in this article.

4.      Could the authors provide an explanation for the differing profiles of scavenging ability observed in the hydrolyzed samples for the ABTS and DPPH radicals?

5.      Minor comments:

(1)   Some hyphens in the text must be removed.

(2)   There is a difference in font size in lines 63-65.

(3)   In Table 1, the meaning of 'p-value (time)' is unclear and needs clarification.

(4)   In Table 2, please use decimal points instead of commas in the numbers.

Author Response

Dear Reviewer,

First of all we want to thank you for your comments which are very well-aimed. 

  1. The section title 'Results' should be revised to 'Results and Discussion.'

    We corrected.

  1. It is noted that certain amino acids (tyrosine, phenylalanine, leucine, valine, and isoleucine), known for their high antioxidant activity, decreased with increased hydrolysis time. However, the antioxidant activity actually increased with extended hydrolysis time. The authors should discuss this apparent contradiction in the Results and Discussion section.

Antioxidant activity may have increased with longer hydrolysis time due to. We can assume that bioactive peptides can also be formed from free amino acids, connecting them to specific sequences. Some amino acids, particularly sulfur-containing amino acids like cysteine ​​and methionine, can act as antioxidants themselves, so the resulting peptide sequences with these amino acids in extremely short chains can have an excellent antioxidant effect.

  1. The differences in results obtained when using pepsin and papain, as well as between liver and heart tissues, should be summarized in this article. We have added. Marking changed places in the manuscript in yellow.
  2. Could the authors provide an explanation for the differing profiles of scavenging ability observed in the hydrolyzed samples for the ABTS and DPPH radicals?

Different methods, reagents, certain measuring equipment always affect the test results when evaluating the same parameter of the same sample. Differences in the results can be assumed to be due to differences in the accuracy of the methods under internal reproducibility conditions, the research was performed in a non-accredited scientific laboratory. Color reactions are very sensitive to inaccuracies, so in order to ensure the quality of the scavenging ability results, measurements and evaluation were carried out using two different methods.

  1. Minor comments:

(1)   Some hyphens in the text must be removed.

we corrected the words: hy drolysates, enzymes, kurie turėtų būti be tarpelių

(2)   There is a difference in font size in lines 63-65.

We corrected.

(3)   In Table 1, the meaning of 'p-value (time)' is unclear and needs clarification.

We corrected.  

(4)   In Table 2, please use decimal points instead of commas in the numbers.

We corrected.  

Round 2

Reviewer 2 Report

Comments and Suggestions for Authors

 My previous comments are not addressed, such as:

1. FRAP is a very important antioxidant measurement technique, it needs to be performed  

2.  A schematic diagram of the experimental process needs to be included in the revised MS

3.  Why MIC is not done?

4.  PCA is required for statistical analysis

5.  Pics for samples need to be included in the revised MS

Comments on the Quality of English Language

Minor editing of English language required

Author Response

Dear Reviewer,
Thank you for your comments. In consideration, we have added the missing information in our manuscript. We highlighted it in yellow.
Best regards, authors

Round 3

Reviewer 2 Report

Comments and Suggestions for Authors

The quality of the MS has been improved. 

Comments on the Quality of English Language

 Minor editing of English language required